# Differential Deep Convolutional Neural Network Model for Brain Tumor Classification

**DOI:** 10.3390/brainsci11030352

**Published:** 2021-03-10

**Authors:** Isselmou Abd El Kader, Guizhi Xu, Zhang Shuai, Sani Saminu, Imran Javaid, Isah Salim Ahmad

**Affiliations:** State Key Laboratory of Reliability and Intelligence of Electrical Equipment, Hebei University of Technology, Tianjin 300130, China; zs@hebut.edu.cn (Z.S.); sansamk4@gmail.com (S.S.); imranpolitely111@gmail.com (I.J.); isahsalimahmad@gmail.com (I.S.A.)

**Keywords:** MRI images, classification, brain tumor, differential deep-CNN, accuracy, loss values

## Abstract

The classification of brain tumors is a difficult task in the field of medical image analysis. Improving algorithms and machine learning technology helps radiologists to easily diagnose the tumor without surgical intervention. In recent years, deep learning techniques have made excellent progress in the field of medical image processing and analysis. However, there are many difficulties in classifying brain tumors using magnetic resonance imaging; first, the difficulty of brain structure and the intertwining of tissues in it; and secondly, the difficulty of classifying brain tumors due to the high density nature of the brain. We propose a differential deep convolutional neural network model (differential deep-CNN) to classify different types of brain tumor, including abnormal and normal magnetic resonance (MR) images. Using differential operators in the differential deep-CNN architecture, we derived the additional differential feature maps in the original CNN feature maps. The derivation process led to an improvement in the performance of the proposed approach in accordance with the results of the evaluation parameters used. The advantage of the differential deep-CNN model is an analysis of a pixel directional pattern of images using contrast calculations and its high ability to classify a large database of images with high accuracy and without technical problems. Therefore, the proposed approach gives an excellent overall performance. To test and train the performance of this model, we used a dataset consisting of 25,000 brain magnetic resonance imaging (MRI) images, which includes abnormal and normal images. The experimental results showed that the proposed model achieved an accuracy of 99.25%. This study demonstrates that the proposed differential deep-CNN model can be used to facilitate the automatic classification of brain tumors.

## 1. Introduction

Brain tumor is one of the most common types of cancer deaths around the world, according to the latest statistics of the World Health Organization. The early diagnosis of a brain tumor saves the patient from death and helps treat patients on time, but this is not always available to people. Gliomas can be considered the most dangerous of the primary brain tumors in the central nervous system (CNS) [1]. In recent years, the World Health Organization, within its revised fourth edition publication in 2016, adopted the existence of two types of gliomas tumors [2] the low-grade (LG) and the high-grade (HG) glioblastomas. The low-grade gliomas will, in general, show benevolent propensities.

Nevertheless, they have a uniform repeat rate and can increase in evaluation over the long run. The high-grade gliomas are undifferentiated [3]. They also convey a more negative prognostic. The most widely recognized strategy for the differential diagnostics of tumor type is the magnetic resonance imaging (MRI).

In the medical field, there is a noticeable increase in the volume of data, and traditional models cannot manage it efficiently. Continuous challenges in the field of medical image analysis are the storage and analysis of large medical data. Nowadays, big data techniques play a very important role in the analysis of medical image data using machine learning. Early tumor location generally relies upon the experience of the radiologist [4]. Usually, biopsy is performed to identify whether the tissue is benign or malignant. Unlike tumors found elsewhere in the body, the biopsy of a brain tumor is not obtained before the end-of-the-brain surgery is performed. The biopsy helps to obtain high-quality images of the complex brain tissue and helps in the diagnosis more accurately [5]. To obtain an accurate diagnostic and evade a medical procedure and subjectivity, it is critical to building a viable diagnostics tool for tumor classification and segmentation from MRI images [4].

The development of new technologies, especially machine learning and artificial intelligence, greatly impacts the medical field, as it has provided an important support tool for medical departments, including medical imaging. Various automated learning approaches are applied to classification and segmentation to process MRI images to support the radiologist’s decision. The supervised approach to classify a brain tumor requires specific expertise to extract optimal features and selection techniques, despite the great potential of this approach [6].

In recent years, unsupervised approaches [7] have obtained researchers’ interest not only for their excellent performances but also because of the automatically generated features, reducing the error rate. Recently, deep learning (DL)-based models have emerged as one of the essential methods for medical image analysis, such as reconstruction [8], segmentation [9], and even classification [10]. Milica et al. [5] proposed a new CNN architecture for brain tumor classification using three types of tumor. They are based on a simple developed network, which already has an existing pre-trained network, and is tested using MRI-T1-weighted images. The model’s overall performance was evaluated using four current methods: combined between two databases and two 10-fold cross-validation methods. The overall capability of the model was tested through an augmented image database. The 10-fold cross-validation model’s best result was achieved for the record-wise cross-validation for an augmented database and the accuracy was 96.56%.

Hiba Mzough et al. [11] presented an efficient and fully automatic deep multi-scale three-dimensional neural network (3D CNN) model for glioma brain tumor classification into low-grade glioma (LGG) and high-grade glioma (HGG) by whole volumetric T1-Gado magnetic resonance (MR) sequence’s that Stand on the 3D convolutional layer and deep network via small kernel. Their model has the potential to merge both the local and global contextual information with reduced weights. They proposed a preprocessing technique based on intensity normalization and adaptive contrast enhancement of MRI data to control the heterogeneity. Moreover, to successfully train such a deep 3D network, they utilized a data augmentation technique. Their work studied the impact of the proposed preprocessing and data augmentation on classification accuracy. Quantitative evaluations, over the well-known benchmark (Brats, 2018), authenticate that their proposed architecture generates the most particular feature map to differentiate between LG and HG gliomas compared with the 2D CNN variant. Their proposed model offers good results which outperform those of other models. The lately supervised and unsupervised state-of-the-art model achieved an overall accuracy of 96.49% using the validation dataset.

Abhishta Bhandari et al. [12] proposed an automatic segmentation model for brain tumor segmentation using MR brain images. Approaches such as convolutional neural networks (CNNs), which are machine learning pipeline methods on the biological process of neurons (called nodes) and synapses (connections), have been of interest in the literature. They investigate the part of CNNs to classify brain tumors by first taking an educational look at CNNs and execute a literature search to regulate an example pipeline for classification. Then, they investigate the future using CNNs by traversing a novel field—radionics. This inspects quantitative features of brain tumors such as signal intensity shape and texture to forecast clinical outcomes such as response to therapy and survival. Linmin Pei [13] proposed a deep learning method for brain tumor classification and overall constancy prediction based on structural multimodal magnetic resonance images (mMRIs). They first suggested a 3D context-aware deep learning that considers tumor area uncertainty in the radiology mMRI image sub-zones to perform tumor classification. They then appeal a regular 3D convolutional neural network (CNN) on the tumor classification to reach the tumor subtype. They perform survival prediction using a hybrid method of deep learning and machine learning. They applied the proposed approaches to the Multimodal Brain Tumor Segmentation Challenge 2019 (BraTS 2019) dataset for tumor segmentation and overall survival prediction and the Computational Precision Medicine database Radiology–Pathology (CPMRadPath) Challenge on Brain Tumor Classification 2019 for tumor classification. The performance evaluation of the model using very famous metrics MSE, accuracy, and dice overall values. Their proposed model results give a robust segmentation, and the classification results ranked in second place at the testing step.

Ahmet Çinar et al. [14] proposed a hybrid method for brain tumor classification using MR images. They used the CNN model based on Resnet50 architecture. They removed the last layers of the Resnet50 model and added eight layers. Their model obtained perfect accuracy. Results are achieved with GoogleNet models, Alexnet, Resnet50, Densenet201, and InceptionV3. Mesut Toğaçar [15] suggests a new convolution neural network model for brain tumor classification named “BrainMRNet.” The architecture is built on attention modules, and a hyperactive column technique with a residual network—their model starts by pre-processing in BrainMRNet. Then, the transfer to the attention model based on augmentation techniques for every image. The model selected an important area of the MR image and transferred it to convolution neural network layers. The BrainMRNet model based on the essential technique in convolution layers is a hyperactive column. The feature extracted from every layer of the brain model is kept by the last layer’s array structure based on the method. The objective is to select the greatest and the most efficient features among the features maintained in the array. The classification success obtained with the BrainMRNet model was 96.05%.

Fatih Özyurt et al. [16] presented a study based on a hybrid model using a neutrosophy co-evolution neural network (NS-CNN). The objective is to classify the tumor area from brain images as benign and malignant. In the first stage, MRI images were segmented using the neutrosophy set—expert maximum fuzzy-sure entropy (NS-EMFSE) method. They obtained CNN and classified using support vector machine (SVM) and k-nearest neighbors (KNN) classifiers using the segmented brain images in the classification stage. The model’s evaluation based on 5-fold cross-validation on 80 benign tumors and 80 malign tumors. The result of the model shows that the CNN features displayed an excellent performance with different classifiers. Javaria Amina et al. [17] created a model based on the DWT fusion of MRI sequences using a convolutional neural network for brain tumor classification. The work proposed a hybrid between four MRI sequences’ structural and texture information (T1C, T1, FLAIR, and T2). Combining a discrete wavelet transform (DWT) with Daubechies wavelet kernel is used for the fusion process, which gives a more informative tumor area as compared to a single individual sequence of MRI. Then, they applied a partial differential diffusion filter (PDDF) to remove the noise. They used a thresholding method to feed the proposed model convolutional neural network (CNN) to classify tumor or non-tumor areas. The authors used five databases—BRATS 2012, BRATS 2013, BRATS 2015, BRATS—and BRATS 2018 to train and test the models.

Pim Moeskops et al. [18] proposed an automatic model for the classification of MR brain images using a convolution neural network. They used multiple patch sizes and multiple convolution kernel sizes to obtain multi-scale information about each voxel. The approach is not dependent on explicit features but learns to recognize the important information for the classification using training data. Their approach wants a single anatomical MR image only, to validate the model using five different datasets that include axial T1-weight and T2-weight images. The model was evaluated using average dice coefficient overall segmented tissue classes for each data set. Jamshid Sourati et al. [19] proposed a novel active learning method based on Fisher information (FI) for CNNs for the first time. The efficient backpropagation approach for computing gradients and a new low-dimensional approximation of FI enabled us to compute FI for CNNs with a large number of parameters. They evaluated the proposed approach for brain extraction with a patch-wise segmentation CNN model in two different learning types: universal active learning and active semi-automatic segmentation. In two scenarios, a starting model was achieved based on the labelled training subjects of a data set. The objective was to gloss a small subset of novel samples to build a model that effectively performs the target subject(s). The dataset includes MR images that differed from the source data through different ages (e.g., newborns with different image contrast). The result of the FI-based AL model showed excellent performance.

Benjamin Thyreau et al. [20] improved a cortical parcellation approach for MR brain images using convolutional neural networks (ConvNets). A machine learning approach which automatically transmits the knowledge achieved from surface analyses onto something immediately viable on simpler volume data. They train a ConvNets model on a big data set of double MRI cohorts’ cortical ribbons to reduplicate parcellation acquired from a surface approach. They forced the model to generalize to unseen segmentations to make the model applicable in a broader context. The model is estimated on the unseen data of unseen cohorts. They described the approach’s behavior during learning. They quantified its credence on the database itself, which resort to granting support for the requirement of a big training database, augmentation, and double contrasts. General, the ConvNets approach provides an efficient method for segmenting MRI images quickly and accurately.

Jude Hemanth et al. [21] proposed a model to classify an abnormal tumor using a modified deep convolution neural network (DCNN). Their aspect is targeted to reduce the computational complexity of conventional DCNN. Favorable modifications are perfect in the training model to reduce the number of parameter amendments. The weight amendment process in the fully connected layer is wholly discarded in the proposed modified method. In-state, a simple task process is utilized to discover the weights of this fully connected layer. Thus, computational complication is safely minimized in the proposed method. The modified DCNN was used to explore magnetic resonance brain images. The performance of the model showed excellent accuracy.

XinyuZhou et al. [22] proposed a model for brain tumor segmentation using efficient 3D residual neural network (ERV-Net), which has less GPU memory consumption and computational complexity. Considering that the ERV-Net is efficient in computation. Firstly, the used ShuffleNetV2 is an encoder to develop the efficiency of ERV-Net and reduce GPU memory, then to prevent degradation, they input the decoder with residual blocks (Res-decoder). For solving the problems of network convergence and data imbalance, they improved a fusion loss function which consists of dice loss and cross-entropy loss. After that, they suggested a concise and robust pre-post approach to improve the coarse segmentation output of ERV-Net. To evaluate their model, they used the dataset of a multimodal brain tumor segmentation challenge 2018 (BRATS 2018) and demonstrated that ERV-Net obtained the best performance with dice of 81.8, 91.21 and 86.62% and Hausdorff distances of 2.70, 3.88 and 6.79 mm for enhancing tumors.

Muhammad Attique Khan et al. [23] proposed an automated multi-model classification approach-based deep learning for brain tumor classification. Their purposed model included five phases. Firstly, they employed the linear contrast stretching based on edge histogram equalization and discrete cosine transform (DCT). Secondly, they performed deep learning feature extraction. By using transfer learning, two pre-trained convolutional neural network (CNN) models, namely VGG16 and VGG19, were used for feature extraction. Thirdly, they implemented a correntropy-based joint learning method with the extreme learning machine (ELM) for its best chosen features. Fourthly, they merged the robust covariant features into one matrix based on the partial least square (PLS) for classification, and the combined matrix was fed to the ELM. Their proposed model was validated using the BRATS database and achieved a best accuracy of 97.8%. This work was motivated by the existing approach [23]. In this approach, multi-modal classification using deep learning was employed for brain tumor classification. The proposed method in this work used a differential deep neural network for brain tumor classification.

This paper proposed a differential deep convolutional neural network model for normal and abnormal MR brain image classification. By using pre-defined hyperactive values and a differential factor, feature maps were generated in differential CNN networks. The differential CNN utilized more differential features maps to extract more detail in the brain MRI image without increasing the number of convolution layers and parameters. For calculating the difference between the pixel and the pixel on the corresponding position of an adjacent layer, we added another fixed filter. The relevant back and the relevant differential feature map processing on the genuine algorithm improved the brain tumor classification performance. The differential deep-CNN model decreased the complexity of convolutional network structures without compromising the values and conformed to the requirements of the computing techniques. The proposed model experiments were performed using an MR database obtained from the Tianjin Universal Center of Medical Imaging and Diagnostic (TUCMD) in real-time. The proposed differential deep-CNN model’s performance was analyzed using the terms of accuracy, sensitivity, specificity, precision, and F-score values. This study covered the problem of low accuracy using deep learning models in clinical application and contributed the reduced complexity of convolution neural network structures without increasing the parameters. The rest of the paper is organized as follows. Section 2 introduces the methodology of the proposed differential DCNN model for a brain tumor. The data collection and augmentation are given in Section 3. The experiment results are given in Section 4. The discussion is given in Section 5. The conclusion and future works are given in Section 6.

## 2. Methodology

### 2.1. Deep Convolutional Neural Network

CNN gives a high-speed and accurate algorithm displaying excellent performance in the detection and classification compared to ancient neural networks [24,25]. CNNs have further improved the classification accuracy of many standard image databases while applying them to solve imaging problems related to vision, such as MNIST [26] and CIFAR 10 [27].

#### 2.1.1. Convolution Layer

The basic architecture of a CNN consists of several convolutional layers, pooling layers, and completely fully connected layers. [28,29]. The function of the convolution layer is to identify the local connection features of the existing layer. The formula for computing a single output matrix is described as follows in Equation (1):(1)$$A_{j}=f\left(\sum_{i=1}^{N}I_{i}*K_{i,j}+B_{j}\right)$$
where *I* is an input vector, and *K* is the corresponding convolution kernel with the size of *B_j_* × *n* (*n* < input size). After that, all the convoluted matrices are added up. A bias value $B_{j}$ is added to each element of the resulting matrix. *f* is a non-linear activation function working on each element of the previous matrix to produce the output matrix *A*.

#### 2.1.2. Activation Function

In this work, a polished linear function was selected as the activation function to assess the CNN’s classification performance and learning speed (Matsuda, Hoashi et al. 2012). The formula in Equation (2) is defined as follows:(2)f(x)=max(0,x)(ReLU)

The pooling layer is responsible for integrating linguistically similar features to reduce the fidelity of feature maps [30].

#### 2.1.3. Back Propagation

It can train the weights of all feature maps and update the training weights [31].

Different structures can affect the training and testing performance in designing CNN models [32]. Usually, the deep learning network performs well, but it takes a longer amount of time, and it can also achieve an external network with very high computational efficiency. However, the problem is often a lack of equipment. The structure of the original CNN is shown in Table 1.

The proposed differential deep CNN model consists of five convolutional layers and five average pooling layers between the convolution layers. Since the differential deep-CNN model aims to classify six types of MR brain images, the last fully connected layer has seven channels. Previous researchers [20,21,24,25,26,27,28,29,30,31,32,33,34] showed that an eminent decrease in the fully corresponding layer size in a CNN would not minimize the network performance.

The experiment’s general objective is to compare the probability of brain tumor classification on the final output. The utmost probable class is considered the last prediction class. The output is the final predictive classification.

### 2.2. Differential Deep Convolutional Feature Map

The key to the deep learning architecture is convolution, in which multiple filters hover over the input images. The feature of the input image is extracted by simulating human vision. However, when increasing the number of feature maps included in the structure’s feature extraction layers, more features are classified.

In classical convolution neural networks, feature maps are created via transferred knowledge or random initialization. The feature maps in differential deep CNNs are produced utilizing classical convolution feature maps by applying pre-defined hyperactive values and a differential operator [35]. We used differential convolution maps to analyze the directional patterns of pixels and their neighborhoods through calculating addition variation. In mathematical differentiation, the sequence change is considered by calculating the difference between the pixel activations. Each feature map is used to calculate the difference in one direction, as shown in Figure 1.

Every feature map is utilized for counting the difference in one direction. From here, additional feature maps are obtained that contain variations in different directions. In contrast [36], we added one static filter to the original algorithm to extract more brain tumor classification task features. Since we added a fixed filter here, feature maps were added directly correspondingly. We let the initial feature map created from conventional neural networks be *f*_1_, and the five feature maps resulting using the differential operator were *f*_2_, *f*_3_, *f*_4_, *f*_5_ and *f*_6_. We calculated the neurons in these maps using the Equations (3)–(7):(3)$$f_{2,i,j}=f_{1,i,j}-f_{1,i+1,j}$$
(4)$$f_{3,i,j}=f_{1,i,j}-f_{1,i,j+1}$$
(5)$$f_{4,i,j}=f_{1,i,j}-f_{1,i+1,j+1}$$
(6)$$f_{5,i,j}=f_{1,i+1,j}-f_{1,i,j+1}$$
(7)$$f_{6,i,j}=f_{1,i+1,j+1}-f_{1,i,j+1}$$
where *i* and *j* are the coordinates of the neurons in the convolutional feature maps. Suppose that the size of *f*_1_ is *M* × *N* and the sizes of *f*_2_, *f*_3_, *f*_4_, *f*_5_ and *f*_6_ are (*M* − 1) × *N*, *M* × (*N* − 1), (*M* − 1) × (*N* − 1), (*M* − 1) × (*N* − 1), (*M* − 1) × (*N* − 1), respectively.

The differential convolutional feature maps are calculated from the first feature map using differential operators after the traditional convolution feature map generates the first feature map. The differential convolution feature maps are utilized to detect an image’s basic features, such as corners and edges.

Based on the derivation process above, we noted that deep differential CNN uses more differential feature maps to extract more details in the images without increasing the convolution layers. Therefore, the proposed differential deep CNN reduces convolution network structures’ complexity, thus decreasing the computing requirements.

### 2.3. Back Propagation

The backpropagation (BP) algorithm is improved while changing feature maps. Suppose the network cannot determine the expected output value in the output layer. In that case, the sum of the error between taking the expected value and the output value as a positional function is moved in the opposite direction. Then, compute the partial derivative of the target function layer by layer. The partial derivative is considered the learning gradient. Modified CNNs are the weights of feature maps based on the learning rate and gradient. When the error is below the expected values, the training stops.

The error transmitted to the first map is *d*_1_; the errors transmitted to the created extra maps and the error matrix are *E*. Equations (8)–(10) illustrate the error calculations for the relevant filter:(8)$$E_{i,j}=d_{1,i,j}-d_{2,i,j-1}+d_{2,i,j}-d_{3,i-1,j}+d_{3,i,j}-d_{4,i-1,j-1}+d_{4,i,j}+d_{5,i-1,j}+d_{5,i,j-1}-d_{6,i,j-1}+d_{6,i-1,j-1}$$

If 1 < *i* < *M* and 1 < *j* < *N*, the *E_i,j_* Equation (8) describes the neurons’ error neither at the edges nor in the corners. It receives error feedback from all neighboring neurons:(9)$$E_{i,j}=\left\{ \begin{array}{l} d_{1,i,j}+d_{2,i,j}+d_{3,i,j}+d_{4,i,j},i=1,j=1\\ d_{1,i,j}-d_{2,i,j-1}+d_{3,i,j}+d_{5,i,j-1},i=1,j=N\\ d_{1,i,j}+d_{2,i,j}-d_{3,i-1,j}-d_{5,i-1,j},i=M,j=1\\ d_{1,i,j}-d_{2,i,j-1}-d_{3,i-1,j}-d_{6,i-1,j-1},i=M,j=N \end{array}\right.$$

The *E_i,j_* in Equation (9) describes the error of the neurons in the concerns and edges. It receives error feedback from three neighboring neurons:(10)$$E_{i,j}=\left\{ \begin{array}{l} d_{1,i,j}-d_{2,i,j-1}+d_{3,i,j}+d_{4,i,j}+d_{5,i,j-1},i=1,1\leq j\leq N\\ d_{1,i,j}-d_{2,i,j-1}+d_{2,i,j}-d_{3,i-1,j}-d_{4,i-1,j-1}-d_{6,i-1,j},i=M,1\leq j\leq N\\ d_{1,i,j}+d_{2,i,j}+d_{3,i,j}-d_{1,i-1,j}+d_{4,i,j}-d_{5,i-j},1\leq i\leq M,j=N\\ d_{1,i,j}-d_{2,i,j-1}+d_{3,i,j}-d_{4,i,j-1,j}+d_{6,i,j-1,}1\leq i\leq M,j=M \end{array}\right.$$

The *E_i,j_* in Equation (10) describes the error of the neurons propagated to the edge neurons. It receives error feedback from 5 neighboring neurons.

## 3. Databases

### 3.1. Database Collection

Normal and abnormal MR brain images collected from Tianjin Universal Center of Medical Imaging and Diagnostic (TUCMD) were used in this paper. Abnormal MR brain images were taken of six abnormal tumor types such as metastasis, meningioma, glioma, astrocytoma, germ cell and craniopharyngiomas. All MR brain images were converted into grey images with a size of 256 × 256 pixels so that there was only one input channel for the differential deep-CNN. This study collected 17,600 MR brain images, which includes T1, T2, and FLAIR images. They were collected by a Philips ingenia 3.0T MR Scanner, Tianjin Philips-Middle Ring Electronic Co.Ltd, Tianjin city, china. The samples are shown in Figure 2a,b.

### 3.2. Database Augmentation

It is possible to ignore the common features of brain MRIs. Due to this, the generalization capacity of the differential deep-CNN model will be impaired. To solve this problem, several data augmentation approaches were proposed to prevent overfitting in this paper; after completing all pre-processing stages and database augmentation for the original database, 25,000 MR brain images were obtained, which included abnormal and normal images. In this TUCMD database, there were 7000 MR brain images for each class of normal images and 9000 MR brain images for the abnormal images. We selected 5-fold validation as the training step. This means every fold consists of 1400 images for normal images and 1800 images for abnormal images.

## 4. Experiments Results

Although DCNN has a strong advantage in representing acquired features, deep structure and supervised learning may cause overproduction when the amount of training data is limited, as is the case with many medical situations in the case of limited data, as a large number of parameters in the differential deep-CNN may lead to over-fitting.

The differential deep-CNN utilized the same values as the original CNN structure in this experiment. They have the same positions and number of the convolution layers and pooling layers. In this differential deep convolution neural network, there are five convolutions layers, and the number of feature maps is 48, 20, 20, 8, and 4. The size of the feature maps set in the first two convolutions layers is 2 × 2, and the size of the feature maps in other convolutions layers is 3 × 3. All convolutional layers follow an intermediate pool layer.

In this work, we ran this simulation using a ThinkStation P620 Tower Workstation, NVIDIA Quadro^®^ P2200 16 GB, Lenovo Company, Tianjin city, China. To create a CNN architecture, we used Tensorflow and Keras, Spyder3.7. We did the training and testing using our collected TUCMD database images to evaluate and analyze different factors in the differential deep-CNN model.

Since the original feature maps of the differential D-CNN are convoluted by the pre-set filter, a single depth expands the convolution layers without increasing the number of convolution layers. Moreover, the differential feature maps register variations in various directions, which improves the performance of the differential deep CNN to identify the basic temple image and the more accurate classification of images. From here, the classification accuracy is improved through using the differential deep-CNN model.

The accuracy and loss curves shown in Figure 3 are standard indexes used to evaluate data sets’ learning performance. Figure 3a represents the accuracy value of the differential deep-CNN model (training and testing steps). The graph shows the effectiveness of the proposed model. The accuracy achieved by the differential deep-CNN model was 0.9925.

Figure 3b represents the loss value (training and testing), the validation loss is the same metric as the training loss, but it is not utilized to update the weights. It is calculated in the same way—by running the network forward over inputs *x_i_* and comparing the network outputs ${\hat{y}}_{i}$ with the ground truth-values using a loss function (11):(11)$$J=\frac{1}{N}\sum_{i=1}^{N}\zeta ({\hat{y}}_{i},y_{i})$$
where *ζ* is the individual loss function somehow based on the difference between predicted value targets. Graph (b) shows the proposed model robustness using a loss value, and the differential deep-CNN model obtained the best loss value (0.1 train and 0.2 test).

In this manuscript, a differential deep-CNN model was deployed to classify the input MR images as abnormal (brain tumor: yes) or normal (brain tumor: no). Figure 4a represents the classification of the abnormal MR brain database based on our proposed model; the results demonstrate the model’s ability to classify big data while maintaining accuracy. Figure 4b describes the classification of normal MR brain images based on the differential deep-CNN model; the results showed the proposed model’s efficacy in classifying the big data of MR brain images.

To evaluate the proposed model’s efficiency, for comparison, we relied on eight models of machine learning methods, between the classical and the modern ones. The comparison between the proposed differential deep-CNN model and the other models was based on the following values:(12)$$Accuracy=\frac{\left(TP+TN\right)}{\left(TP+TN)+(FP+FN\right)}\times 100$$
(13)$$Sensitivity=\frac{\left(TP\right)}{\left(TP+FN\right)}\times 100$$
(14)$$Specificity=\frac{\left(TN\right)}{\left(TN+FP\right)}\times 100$$
(15)$$\mathrm{Pr}ecision=\frac{\left(TP\right)}{\left(TP+FP\right)}\times 100$$
(16)$$F-Score=\frac{2xTP}{2xTP+FP+FN}\times 100$$
Where true positive (*TP*), true negative (*TN*), false positive (*FP*), and false negative (*FN*) are used to evaluate the performance of the proposed model and eight machine learning models for comparison purposes.

### 4.1. K-Nearest Neighbors Model (KNN)

It is a non-parametric machine learning model; it is used for image classification. In both cases, the input consists of the closest training examples in the feature space. The output depends on whether k-NN is used for image classification [37].

### 4.2. Convolutional Neural Network with the Supper-Vector Machine Model (CNN-SVM)

A convolutional neural network is a type of deep neural network, most commonly applied to analyze visual images. The supper vector machine supervises learning methods with associated learning algorithms that analyze data for image classification. SVMs are one of the most robust prediction approaches based on statistical learning frameworks [38].

### 4.3. Traditional Convolutional Neural Network Model (CNN)

The CNN algorithm achieved excellent results in MR image segmentation and classification [5].

### 4.4. Modified Deep Convolutional Neural Network (M-CNN)

Deep convolutional neural networks (DCNNs) are widely utilized deep learning networks for medical image analysis. Generally, a D-CNN’s accuracy is very high, and in these networks, the manual feature extraction process is not necessary. The high accuracy adds huge computational complexity. Appropriate modifications have been made in the training algorithm to reduce the number of parameter adjustments, and reduce the computational complexity of conventional DCNN [21].

### 4.5. Alex- Net, GoogleNet, and VGG-16 Models

Alexnet, GoogleNet, and VGG-16 are names as a convolutional neural network (CNN). The overall CNN structure consists of convolutional layers, fully connected layers, and pooling layers [39,40].

### 4.6. BrainMRNet Model

The BrainMRNet model consists of some of the CNN structure layers (convolution, pooling, fully). The dense layer was utilized before the classification step. The dense layer is a type of hidden layer. The dense layer loads the values of a matrix vector and is updated continuously during the backpropagation. With the dense layer, the matrix size where the values are preserved is changed [41]. Among the most important techniques are the BrainMRNet model, which is utilized as a hyperactive column in the convolutional layers [15].

This paper compared eight popular models: KNN, CNN-SVM, CNN, Modified Deep-CNN, Alex-Net, Google-Net, VGG-16, and BrainMRNet, with the differential deep convolutional neural network (differential deep-CNN) model. We performed the comparison using accuracy, sensitivity, specificity, precision, and F-score values, the details of which are shown in Table 2.

We can notice in Table 2 that the accuracy, sensitivity, specificity, precision, and F-score of a differential deep-CNN model on a testing and training database accomplished the best results of 99.25, 95.89, 93.75, 97.22, and 95.23%, which is significantly superior compared with previous models.

## 5. Discussion

In the literature survey, there were studies by researchers that used a large classification database with trained networks such as in [42,43,44]. That, as inputs, they used a tumor region of the brain [45,46]. Rehman et al. [44] augmented the data and image processing with contrast enhancement. The augmentation was 5-fold, with horizontal and vertical flipping and rotations of 90, 180, and 270 degrees. The best result was achieved with a fine-tuned VGG16 trained based on the stochastic gradient descent with momentum. Very deep networks such as VGG16 and AlexNet require special hardware to perform in real-time.

Phaye et al. [47] developed a deep learning model for the feature extraction and mixture of experts for classification. For the first step, the outputs of the last max-pooling layer of a convolution neural network (CNN) are utilized to extract the hidden features automatically. For the second step, a mixture of advanced variations of extreme learning machine (ELM) consists of basic ELM, constraint ELM (CELM), on-line sequential ELM (OSELM), and kernel ELM (KELM), is improved. The proposed model achieved an accuracy of 93.68%.

Gumaei et al. [48] proposed a hybrid feature extraction model with a regularized extreme learning machine to improve an accurate brain tumor classification model. The model starts by extracting the features from brain images based on the hybrid feature extraction method; then, it computes the covariance matrix of these features to enterprise them into a new considerable set of features based on principle component analysis (PCA). Then, regularized extreme learning machine (RELM) is utilized for classifying the type of brain tumor. The proposed achieved accuracy was of 94.23%.

Ge et al. [49] proposed a novel multistream deep CNN model for glioma grading by applying sensor fusion from T1-MRI, T2, and FLAIR MR images to enhance performance by feature aggregation; then, they were mitigated through overfitting based on 2D brain image slices in combination with 2D image augmentation. The proposed model obtained an accuracy of 90.87%.

Table 3 shows a comparison between the proposed differential deep-CNN and the existing models. From Table 3, we note that the accuracy of the proposed model is much better than that of the previous model. We can conclude that the proposed model offers powerful methods for accurate brain tumor classification, outperforming several recent CNN models.

The differential deep-CNN model performance was evaluated with twelve previous models, such as [5,15,21,23,37,38,39,40,41,47,48,49]. Through observing the aforementioned experimental results, the performance of the proposed technique is better as compared to previous approaches, which demonstrate the applicability of the proposed model.

In this study, to discuss the threats-to-validity of the experimental results, we used the last references published in the field of brain tumor classification using different deep learning techniques, as shown in Table 2 and Table 3. The validation of our results based on a comparison between existing models results in a simulation with a proposed differential deep-CNN model using five values. The ability of the proposed model to validate UTMC big data with high accuracy and very low loss validation as shown in Figure 3 is clear evidence of the threat-validity of the results.

## 6. Conclusions

CNN deep networks have achieved great success in recent years in analyzing medical images, as a relatively low-cost means of detecting and classifying accuracy in contrast to its expensive computational costs that restrict the application of CNN in clinical practice [50].

A differential deep convolutional neural network was proposed in this work for MR brain image classification. The differential deep-CNN model analyses are based on accuracy, sensitivity, specificity, precision, F-score and loss values. Significant improvement was achieved with 99.25% accuracy, sensitivity 95.89%, specificity 93.75%, precision 97.22%, F-score 95.23% and loss from 0.1 to 0.2 with the proposed model compared with previous models as shown in Table 2 and Table 3. The proposed differential deep-CNN model utilized in the classification obtained better performance than the other methods in the brain tumor classification problem as shown in Figure 4a,b.

Improving deep learning models requires a huge database of medical images. Our scientific team obtained this huge data by TUCMD, which improved our proposed approach’s performance. Furthermore, our differential deep-CNN model uses a database that has been diagnosed, evaluated, and processed by professional medical doctors for evaluation. The experimental results demonstrated that the brain tumor classification of TUCMD data using the proposed model was compatible with the clinicians’ diagnoses. The proposed differential deep-CNN approach shows the importance of deep learning in medicine and the future of deep learning in clinical applications.

In future work, we will examine the overall success of our differential deep-CNN model. We will improve the parameters of the differential filter to make the network coverage faster. We will improve the deep network architectures by adding a multi-channel classifier that improves the classification performance more effectively than before.

## Figures and Tables

**Figure 1 brainsci-11-00352-f001:**
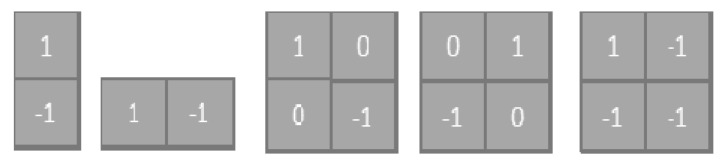
The architecture of the predefined filters.

**Figure 2 brainsci-11-00352-f002:**
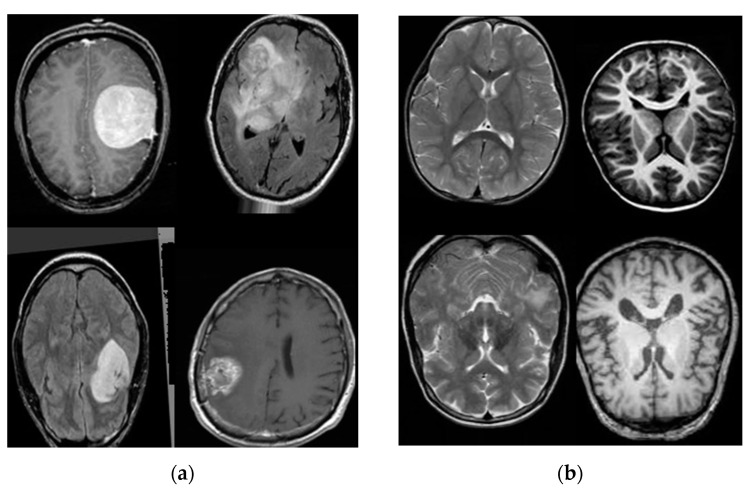
(**a**) Samples of abnormal T1, T2 and FLAIR MR brain images and (**b**) the samples of normal T1, T2 and FLAIR MR brain images.

**Figure 3 brainsci-11-00352-f003:**
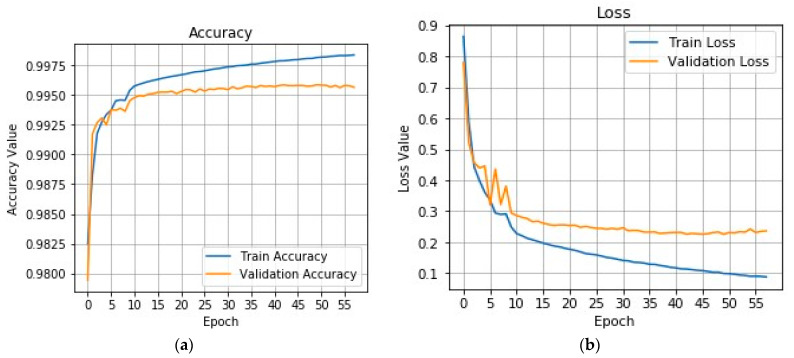
The graph (**a**) represents the results of the accuracy validation value on the Tianjin Universal Center of Medical Imaging and Diagnostic (TUCMD) database obtained by differential deep-CNN model; and the graph (**b**) represents the results of the loss validation value on the TUCMD database achieved by differential deep-CNN model.

**Figure 4 brainsci-11-00352-f004:**
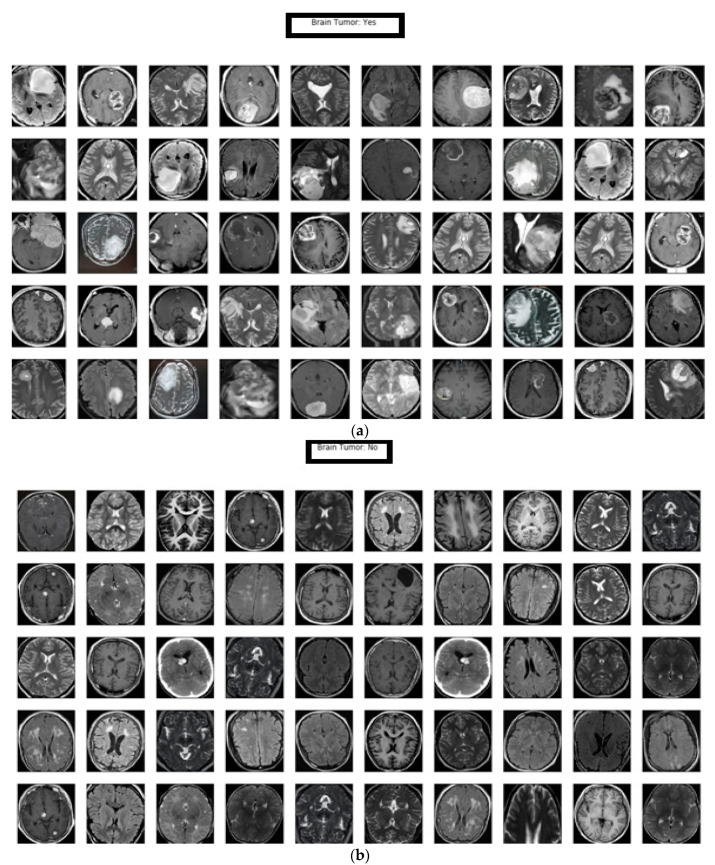
(**a**). Samples represent the results of abnormal T1, T2 and FLAIR MR brain images classification obtained by the differential deep-CNN model and (**b**) the samples represent the results of the normal T1, T2 and FLAIR MR brain images classification achieved by the differential deep-CNN model.

**Table 1 brainsci-11-00352-t001:** Describe the structure of the convolutional neural network (CNN) models model.

Layer	Number of Feature Maps	Kernel Size	Stride	Size of Feature Maps
Input		11		1020 × 1020
Convolution (1)	12	2	2	500 × 500 × 12
Pooling (1)	1	5		250 × 250 × 12
Convolution (2)	5	2	1	250 × 250 × 60
Pooling (2)	1	6		125 × 125 × 60
Convolution (3)	5	3	1	120 × 120 × 300
Pooling (3)	1	3		40 × 40 × 300
Convolution (4)	2	2	1	40 × 40 × 600
Pooling (4)	1	3		20 × 20 × 600
Convolution (5)	1	3	1	18 × 18 × 600
Pooling (5)	1			6 × 6 × 600
F1				21,600

**Table 2 brainsci-11-00352-t002:** Comparison of the results of different models with our proposed differential deep-CNN model.

Model	Accuracy %	Sensitivity %	Specificity %	Precision %	F-Score %
KNN	78	46	50	72.11	68
CNN-SVM	95.62	-	95	92.12	93.11
CNN	96.5	95.07	-	94.81	94.93
M-CNN	96.4	95	93	95.7	94.2
Alex_Net	87.66	84.38	92.31	93.1	88.52
Google-Net	89.66	84.85	96	96.55	90.32
VGG-16	84.48	81.25	88.48	89.66	85.25
BrainMRNet	96.5	95	93	92.3	94.12
Proposed differential deep-CNN	99.25	95.89	93.75	97.22	95.23

**Table 3 brainsci-11-00352-t003:** Comparative differential D-CNN model with CNN-based methods using accuracy.

Model/Year	Data	Model	Accuracy %
Phaye et al. [47]	Accidents Images	CNN	93.68
Gumaei et al. [48]	MRI	ELM-KELM	94.23
Muhammad Attique Khan et al. [23]	MRI BRATS	(ERV-Net)	97.8
Ge et al. [49]	MRI	Multi-stream CNN	90.87
Proposed differential deep-CNN	MRI	Differential D-CNN	99.25

## Data Availability

The data presented in this study are available on request from corresponding author.

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
