# Peer review of "Differential Deep Convolutional Neural Network Model for Brain Tumor Classification"

_brainsci, 2021, doi:10.3390/brainsci11030352_

Round 1

Reviewer 1 Report

The paper proposed a differential deep convolutional neural network (CNN) model to classify different types of brain tumors from the MRI images. The performance of the method is tested on a large dataset of MRI images. The paper requires major revisions.

Comments:

  1. Explicitly state the novelty and contribution of your paper at the end of the first section. What is the knowledge gap closed by this study?
  2. The literature review in the Introduction section is poorly organized and unstructured. Preferably, a separate section should be introduced for literature reviews. The review should be addressed more systematically, by discussing separately different related subdomains such as brain image denoising, feature selection and image segmentation. The following state-of-the-art works are suggested to supplement and improve the quality of your analysis: “Cross-modality deep feature learning for brain tumor segmentation”, “ERV-net: An efficient 3D residual neural network for brain tumor segmentation”, “Multimodal brain tumor classification using deep learning and robust feature selection: A machine learning application for radiologists”. Discuss the limitations of previous works as a motivation for your study. Summarize the overviewed works as a table.
  3. Some subsections in Section 2 and Section 4 are very short – just one sentence. These should be extended or merged with adjacent subsections.
  4. Check the equation (10), I think there should be “<=” instead of “<” in the conditions.
  5. Figure captions should be more informative and descriptive.
  6. How do you deal with the over-fitting problem? What specific architectural features of the proposed neural network allow to manage it?
  7. How many images in the training fold after dataset augmentation was applied?
  8. How did you deal with the class imbalance problem in the dataset? How do you shift the effect of dataset imbalance on the performance evaluation?
  9. Tables 2 and 3 present values in percentages, but equations (11-15) do not calculate percentage values. Use English comma point (.).
  10. Figure 4: present actual brain images instead of screenshots of brain images.
  11. Add the discussion section, and discuss threats-to-validity of the experimental results.
  12. Introduce and explain all abbreviations (such as MR, MRI) on their first occurrence.
  13. Language should be revised. Some statements look strange such as “gliomas could be considered as the most forceful”.
  14. References in the reference list are unnumbered, so they are difficult to follow.

Author Response

The response is attached.

Reviewer 2 Report

The authors presented an original differential deep convolutional neural network model (differential deep-CNN) to classify brain tumors. The author's model obtained an accuracy of 99.25%; the authors, therefore, concluded that the proposed model could be useful to improve the automatic classification in neuro-oncology

Big data and deep/machine learning represent new frontiers in medical sciences, and every new input could help readers and other researchers to deeply and better explore such field. However, I have several issues about the present article, that must be explained.

The abstract (and, in fact, the whole manuscript) must be correct for English and syntax: there are different errors (i.e., lines 10-11 "In the past years. Deep learning techniques have achieved" or lines 32-33 "cancer is one of the highest diseases and deaths in the world".. just a couple of examples, but not the only). Other sentences look without scientific sounds: "Among the central nervous system (CNS) essential brain tumors, gliomas could be considered as the most forceful" (lines 34-35). Does essential mean primary? Also, "forceful", I think is not the right term. 

Lines 36-37: "the World Health Organization, within its revised fourth edition publication 2016, adopted the existence of two types of gliomas tumors: the low grade (LG) and the high-grade (HG) glioblastoma". Whereas this is not the novelty of the 2016 WHO classification of primary brain tumors and the distinction between LGG and HGG dates to previous classifications, glioblastomas are a part of high-grade gliomas.

However, I think that the main limitation is the assumption that "To analyze whether the tissue is benign or malignant, a biopsy is typically performed" This is true for the rest of the body, but usually less for brain tumors: there are a lot of characteristics (i.e. contrast enhancement or not, T1 and T2 signal, etc ) that usually lead to a diagnosis of the presented grading; obviously, the sure diagnosis is obtained with histopathological analysis of surgical samples. This concept must be clarified.

Moreover, the authors should present the introduction as an overview of the current scenario, then introducing the aim of their work. In the current form, the introduction looks like more a discussion section

The results should be modified, and it is hard to understand the value of some subsection (4.1-.4.5) constituted only by a single sentence. 

Author Response

The response is attached.

Round 2

Reviewer 1 Report

The paper has been well revised.

Reviewer 2 Report

The papers has been greatly revised, and it is now more suitable for publication